# Experimental and Computational Studies of Peristaltic Flow in a Duodenal Model

Nadun Palmada [1,2], John E. Cater [3], Leo K. Cheng [1,2] and Vinod Suresh [1,3,*]

1   Auckland Bioengineering Institute, University of Auckland, Auckland 1010, New Zealand; tpal287@aucklanduni.ac.nz (N.P.); l.cheng@auckland.ac.nz (L.K.C.)
2   Riddet Institute, Massey University, Palmerston North 4442, New Zealand
3   Department of Engineering Science, University of Auckland, Auckland 1010, New Zealand; j.cater@auckland.ac.nz
*   Correspondence: v.suresh@auckland.ac.nz; Tel.: +64-9-923-9746

**Abstract:** We study peristaltic flow in a C-shaped compliant tube representing the first section of the small intestine—the duodenum. A benchtop model comprising of a silicone tube filled with a glycerol-water mixture deformed by a rotating roller was created. Particle image velocimetry (PIV) was used to image flow patterns for deformations approximating conditions in the duodenum (contraction amplitude of 34% and wave speed 13 mm/s). Reversed flow was present underneath the roller with fluid moving opposite to the direction of the peristaltic wave propagation. Deformations of the tube were imaged and used to construct a computational fluid dynamics (CFD) model of flow with moving boundaries. The PIV and CFD vorticity and velocity fields were qualitatively similar. The vorticity field was integrated over the imaging region to compute the total circulation and there was on average a 22% difference in the total circulation between the experimental and numerical results. Higher shear rates were observed with water compared to the higher viscosity fluids. This model is a useful tool to study the effect of digesta properties, anatomical variations, and peristaltic contraction patterns on mixing and transport in the duodenum in health and disease.

**Keywords:** gastrointestinal tract; digesta; PIV; CFD; retropulsive flow; bowel; digestion

## 1. Introduction

Peristaltic motion in fluids is induced by the propagation of wavelike contractions along the walls of a flexible tube. In living systems, peristalsis is achieved by the successive contraction and relaxation of muscle layers in the vessel walls. Peristaltic flow is the primary physiological mechanism in tubular organs in the human body, e.g., transport of food through the gastrointestinal (GI) tract, transport of urine through the ureter, and the flow of lymph [1].

Our interest is in understanding the peristaltic flow occurring within the human small intestine, specifically the duodenum. It is the first and shortest segment of the small intestine and takes the form of a C-shaped tube roughly 20 cm in length and 2.5 cm in diameter [2]. In the duodenum, partially digested food (chyme) from the stomach gets well mixed with digestive secretions from the pancreas and liver. The role of the different contraction patterns in the mixing and emptying of chyme within the duodenum remains poorly understood.

Intestinal dysmotility disorders affect more than 15% of the global population and associated with these are considerable health-care costs [3]. Irritable bowel syndrome (IBS) is a common disorder of the small and large intestine that is characterised by abdominal pain and altered bowel habits [4]. Chronic intestinal pseudo-obstruction (CIPO) is an example of a chronic digestive disorder that is characterised by signs of intestinal obstruction without concrete evidence of an actual mechanical obstruction [5]. CIPO is also associated with weak or unsynchronised contractions of the intestinal wall [6]. The impact of these dysmotility

patterns on the flow and mixing of chyme is poorly understood and in vivo measurements of flow within the intestines is very difficult from a technical and ethical perspective.

A limited number of studies have examined peristaltic flows in the GI tract using experiments. A human gastric flow simulator (GFS) was previously developed to study the flow and mixing of gastric contents under peristalsis [7]. The GFS is a rectangular channel with rigid side walls and elastic top and bottom walls made of rubber. Gastric peristalsis is simulated via the controlled deformation of the rubber walls using a set of rollers. Particle image velocimetry (PIV) was used to measure the 2D flow field within the center of the channel using Newtonian liquid digesta (with varying viscosities) and liquid-solid digesta. Laminar flow was measured with a maximum Reynolds number of 125 and a maximum flow velocity of 10 mm s$^{-1}$ was observed in the region of greatest occlusion. Retropulsive flow against the direction of peristalsis was observed with liquid digesta, and changes in the viscosity (1 to 100 mPa·s) had negligible effects on the maximum velocity. This study also included a CFD model of the GFS. Good agreement between the PIV and CFD results was found for the maximum flow velocity in the occluded region.

Another study developed an in vitro model of peristaltic flow in the small intestine by squeezing a silicone elastic tube under a set of rollers [8]. Velocity profiles were measured using an ultrasound velocity profiling (UVP) technique. UVP measures the instantaneous uni-dimensional velocity profile along a measurement line and these measurement lines had to be placed in the section of tubing between the rollers, i.e., the flow profile could not be measured directly underneath the roller. A non-Newtonian shear thinning carboxymethyl-cellulose solution was used to represent the rheological properties of the chyme in the small intestine. A higher speed of peristalsis was found to lead to a higher magnitude of back flow. The main limitation of the UVP technique used in this study was the inability to measure the 2D spatial flow patterns resulting from peristalsis, i.e., an extensive number of 1D measurement lines would be required to evaluate the spatial mixing that occurs as a result of peristaltic contractions.

In silico models of intestinal motility such as computational fluid dynamics (CFD) models provide a better understanding of the fluid flow patterns found within the duodenum compared to existing in vitro simulator studies. CFD models of intestinal motility typically use simplified geometries to represent the small intestine, e.g., 2D rectangular channel [9–11] and straight cylinder [12]. A more realistic C-shaped cylinder has also been used to represent the intestinal geometry [13]. These CFD models have used idealised contraction patterns such as sinusoidal waves [12,13], whereas Love et al. [10] and De Loubens et al. [11] used more realistic contraction patterns derived from ex vivo intestinal samples.

We have developed an anatomically realistic model of intestinal motility using an anatomically realistic 3D geometry of the human duodenum where peristaltic contractions are derived from electrophysiological recordings of slow waves [14]. None of the aforementioned In silico studies (including our previous work) have validated the numerical results. The aim of this study is to validate the computational methods used in the anatomically realistic CFD model. This was done by measuring the tube wall deformations and flow patterns within a C-shaped tube undergoing peristaltic contractions and comparing against a CFD model that uses the same geometry and wall deformations. The paper is organised as follows. First the construction of the physical model, experimental procedures and development of the CFD model are outlined. Then the flow patterns, velocity distributions, and vorticity distributions obtained from PIV and CFD are compared. Finally, flow patterns in a shear thinning power law fluid representing whole digesta are investigated using CFD simulations.

## 2. Materials and Methods

Figure 1 shows a CAD rendering of the bench-top apparatus used to study peristaltic flow within a C-shaped tube. The model consists of flexible transparent silicone tubing (RS PRO 667-8450, inner diameter $D = 8$ mm and 1.6 mm wall thickness) secured around

a 3D printed tube guide with a 0.5 mm deep groove (Figure 1B) to prevent slippage. A roller (20 mm diameter) machined out of acetyl plastic is friction fitted to the tube guide. The tube guide–roller combination attaches onto a Watson Marlow 520S/R2 peristaltic pump. The entire set up is placed in an acrylic enclosure. During operation the tube and the enclosure are filled with fluid. The silicone tubing is fixed onto two hose couples (attached to the left-most edge of acrylic enclosure) and the external ends of the hose couples are connected by tubing to form a closed circuit. Rotation of the peristaltic roller results in an amplitude of peristaltic contraction of 34% (relative to the resting radius), frequency of 3 cycles/min, 40 mm radius of curvature, and wave speed of 13 mm/s, which approximates the contraction amplitude and flow velocity observed in the duodenum [15].

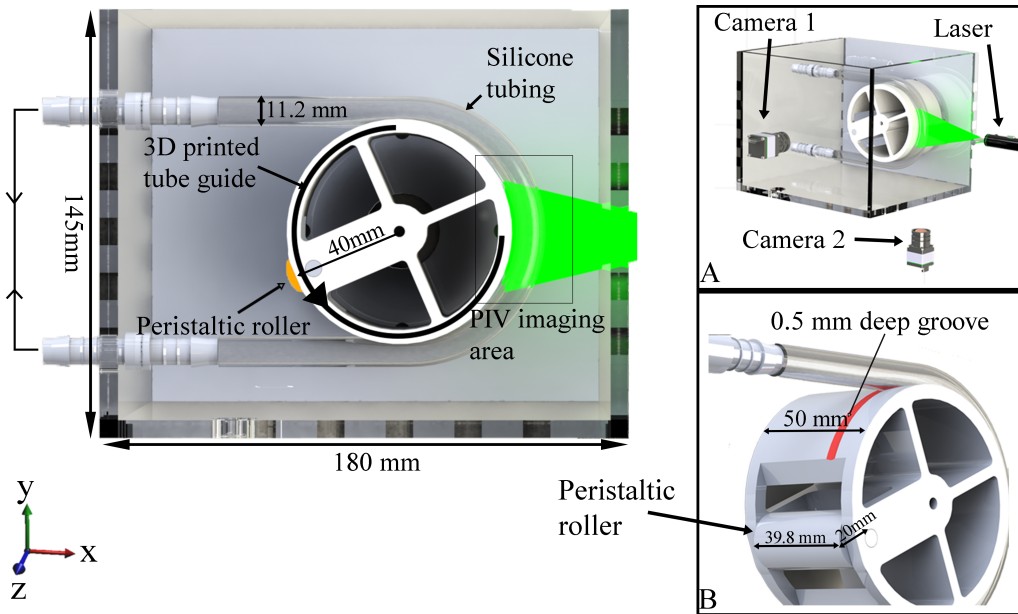

**Figure 1.** Details of the bench-top apparatus. Side panels: (**A**) perspective view showing the arrangement of the cameras, laser source, and resulting laser sheet illuminating the middle right section of tube for PIV; (**B**) rendering of 3D printed tube guide (tube groove highlighted in red) and peristaltic roller.

### 2.1. Model Construction

The refractive index of the tubing was measured to be 1.413 using a refractometer (Hanna Instruments HI-96800) and this was matched using a 62 wt% glycerol-water solution. Refractive index matching was confirmed by monitoring the distortion of grid lines under ambient lighting when imaged through the liquid-filled enclosure and tubing. The glycerol solution is a Newtonian fluid with a dynamic viscosity $\mu = 10.09$ mPa·s and density $\rho_F = 1156$ kg m$^{-3}$ [16]. Silver coated hollow glass spheres (diameter $d = 10$ μm, density $\rho_P = 1040$ kg m$^{-3}$, Dantec Dynamics, catalog # 80A7001) were used as tracer particles. The Stokes number ($St = \rho_p d^2 U / 18\mu D$) and Archimedes number ($Ar = gd^3\rho_F(\rho_P - \rho_F)/\mu^2$) for these particles were both estimated to be on the order of $10^{-7}$, suggesting that the particles closely follow fluid motion and buoyancy effects are negligible. A characteristic flow velocity $U = 0.01$ m/s was used in the calculations based on the peak velocities expected.

### 2.2. PIV Analysis

A laser pointer (130 mW with a wavelength of 532 nm) in combination with a glass rod (6 mm diameter) was used to create a laser sheet along the center of the tube (Figure 1A) to illuminate the tracer particles. These particles were visualised using a FLIR Blackfly S monochromatic (BFS-U3-16S2M-CS) 1.6 MP (1440 × 1080) high speed camera with a frame-rate of 60 fps and the images were directly recorded to a computer. PIV measurement

was performed in the region where the tube is undergoing maximum deformation (on the middle right of the phantom model, as shown in Figure 1).

PIV image processing was conducted using PIVlab 2.53 [17], a GUI based tool in MATLAB (The MathWorks Inc., Natick, MA, USA). The images were masked to remove the roller, tube walls, and other features such that only the tracer particles were visible in the images. Initial image pre-processing included contrast enhancement using contrast limited adaptive histogram equalization. The size of the interrogation area was set to $64 \times 64$ pixels for the first pass of the direct cross-correlation algorithm, with a 50% horizontal and vertical offset of the interrogation windows and $32 \times 32$ pixels for the second pass. Post-processing of the velocity vectors included using a standard deviation threshold filter ($3\sigma$) and a local ($3 \times 3$ pixels neighbourhood) median threshold filter to replace spurious vectors with interpolated ones.

*2.3. Tube Deformation Imaging*

The deformation of the internal tube wall as it undergoes peristaltic contractions was imaged in order to provide the domain for the CFD model. Two different imaging studies were required to obtain a full 3D reconstruction of the internal walls. The first study measured the deformation in the plane of the roller ($x - y$ plane) using camera 1 and the second study involved illuminating the tube walls with a laser sheet to measure the deformation in the plane perpendicular to the roller ($x - z$ plane) using camera 2.

Dark blue food colouring was added to the glycerol and water solution as a contrast agent in order to distinguish the edge between the tube walls and the fluid. The time-series images obtained from the first imaging study were processed with MATLAB using multilevel image thresholding to segment the fluid region from the image. The outer edges of the segmented fluid were extracted to obtain the fluid domain deformation along the center of the tube in the $x - y$ plane. These outer edges were sampled at 25 equidistant points by spline fitting, and a corresponding centerline of each frame was also computed.

The second imaging study consisted of imaging the deformation of the tube in the $x - z$ plane (i.e., expansion of the tube as it contracts in the $x - y$ plane), this study required synchronised capture with two BlackFly high-speed cameras. The tube walls were illuminated using a laser sheet, and camera 1 (BFS-U3-50S5M at 30 fps) imaged the position of the roller (Figure 2A) whilst camera 2 (the same camera used for the PIV study) imaged the deformation of the tube walls in the $x - z$ plane at the illuminated cross-section (Figure 2B). The inner wall of the tube was manually segmented for three different key frames: (i) resting, (ii) tube not in contact with the roller, and (iii) fully deformed (Figure 2B left to right respectively), to obtain cross-sections in the $x - z$ plane.

The corresponding frame from camera one allows the correct key-frame from camera two to be determined based on the roller position. These key-frame cross-sections are placed along the 25 equally sampled points along the centerline and cross-sections were scaled radially to fit two outer edges traced from the imaging study (Figure 2C). Smooth 3D spline interpolation between subsequent points along $x - z$ cross-sections allowed a 3D geometry to be generated for a specific frame. As the roller moved in a counter-clock wise direction in the $x - y$ plane, corresponding cross-sections are placed at the appropriate centerline positions based on the location of the roller. A 3D geometry of the tube inner-walls (i.e., fluid domain boundary) was reconstructed at every 1/30 s.

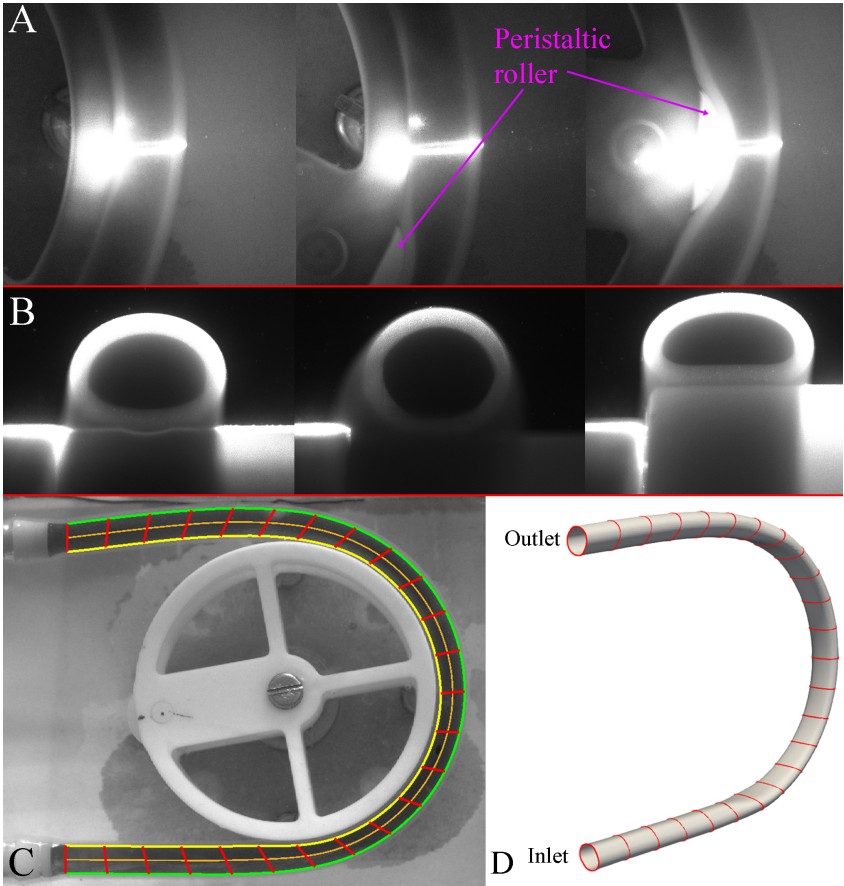

**Figure 2.** 3D geometry reconstruction; (**A**) images obtained from camera 1 during the second deformation study, (**B**) three key cross-sections obtained from camera 2 during the second deformation study, (**C**) deformation of the tube in the $x - y$ plane, resulting edge trace of the fluid (yellow and green lines), computed centerline (orange line) and locations of 25 tubular cross-sections (red lines) and (**D**) reconstructed 3D geometry of the initial fluid domain.

### 2.4. Computational Model

Three dimensional CFD simulations were conducted in the finite volume solver Open-FOAM 5 using the reconstructed tube geometry and deformations to define a moving boundary problem and determine the resulting flow and pressure fields as functions of time. The simulations used the incompressible continuity and Navier-Stokes equations, shown below in the Einstein notation:

$$\frac{\partial u_i}{\partial x_i} = 0, \tag{1}$$

$$\rho \left( \frac{\partial u_i}{\partial t} + u_j \frac{\partial u_i}{\partial x_j} \right) = -\frac{\partial p}{\partial x_i} + \mu \frac{\partial^2 u_i}{\partial x_j \partial x_j}, \tag{2}$$

where $u_i$ is the fluid velocity in direction $x_i$ ($x_1 = x$, $x_2 = y$, $x_3 = z$), $\rho$ and $\mu$ are the fluid density and viscosity, respectively, and $p$ is the fluid pressure.

A no slip condition was imposed at the tube wall ($x_i = x_i^w$) where the fluid velocity is equal to the wall displacement velocity:

$$u_i = \frac{dx_i^w}{dt}, \tag{3}$$

where the wall velocity is determined from the tube deformation imaging. Since the flow is driven by boundary deformations, the magnitude and direction of fluid flow at the

inlet and outlet were not prescribed. Instead, flux values at the inlet and outlet boundary cell nodes were used to determine velocity values at the corresponding boundary faces. If fluid leaves the domain (positive flux in the OpenFOAM convention), a zero velocity gradient condition was imposed: $\frac{\partial u_i}{\partial x_i} = 0$. If fluid enters the domain (negative flux in the OpenFOAM convention), the face velocity was determined from the node velocity component normal to the face.

At the inlet and outlet, the total pressure was set equal to zero, whilst a zero gradient condition for pressure was applied at the wall. The OpenFOAM implementations of the individual boundary conditions are listed in Table 1.

**Table 1.** OpenFOAM boundary Conditions.

| Location | Pressure | Velocity |
|---|---|---|
| Inlet and Outlet | *totalPressure* | *pressureInletOutletVelocity* |
| Wall | *zeroGradient* | *movingWallVelocity* |

The velocity and pressure fields were initialised to be zero. Three cycles (rotations) of the roller were simulated to remove the transient effects of initial conditions. Results from the third cycle are presented in the later sections.

A structured 3D finite volume mesh of the resting tube geometry was generated on cfMesh (an automatic opensource meshing utility) [18], with converged flow solutions requiring a mesh containing approximately one million cells. Mesh convergence was checked by computing the circulation (see Equation (4)) and there was on average a 3% difference (relative to the coarse mesh) in the circulation as the roller moved over the imaging region (Figure 3). The 3D position of the outer surface points in the CFD mesh were computed based on the reconstructed geometries and cubic splines were fitted over time for the individual $x$, $y$ and $z$ coordinates of the CFD surface mesh nodes. These spline coefficients are used by OpenFOAM to compute the location of an individual node at any given time, which allows the flow solver to adjust the timestep based on the Courant ($C_o$) number.

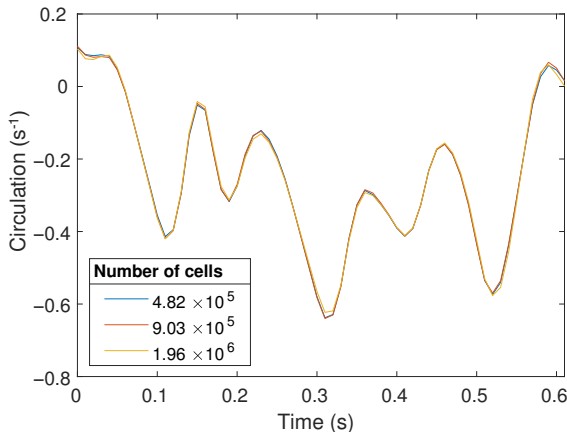

**Figure 3.** Circulation over the PIV imaging region as the roller passes over this region for meshes of different resolutions.

The motion of internal nodes in the CFD mesh are calculated based on the motion of the surface nodes by using the displacement Laplacian solver with an *inversePointDistance* function for the diffusivity, which determines how much the internal mesh points move in relation to the boundary motion. Due to the unsteady deformation of the tubular walls, transient flow simulations were conducted. The transient solver *pimpleDyMFoam* built into OpenFOAM was used to obtain the flow solution. The PIMPLE algorithm combines

both the SIMPLE and PISO algorithms and is typically used for transient problems as stable solutions can be obtained with large Courant numbers, therefore allowing large time-steps [19]. The PIMPLE algorithm settings used for these simulations are shown in Table 2. Adaptive time-stepping was used with a maximum $C_o$ of 3 (spatial average of 0.09), an average time step of 7 ms, and flow solutions were saved every 0.01 s.

**Table 2.** PIMPLE Algorithm Settings.

| | | |
|---|---|---|
| nOuterCorrectors | | 500 |
| nCorrectors | | 2 |
| nNonOrthogonalCorrectors | | 1 |
| correctPhi | | yes |
| residualControl | U-tolerance | $1\times10^{-4}$ |
| | p-tolerance | $1\times10^{-4}$ |

Converged flow solutions required on average 20 iterations (outer-correctors) per timestep with a computation time of 8 h per 20 s cycle using 30 CPU cores (Intel Broadwell E5-2695v4).

In addition to the glycerol solution used in the PIV experiments, CFD simulations were conducted (using the same boundary deformations) for water (Newtonian, $\mu = 8.01 \times 10^{-4}$ Pa·s) and whole digesta (non-Newtonian), to study the effect of fluid properties on the resulting fluid flow patterns from peristalsis. Whole digesta was represented by the power law equation: $\eta = k\dot{\gamma}^{n-1}$, where $\eta$ is the dynamic viscosity, $k$ is the consistency constant ($8.01 \times10^{-4}$ Pa· s$^n$), $n$ is the power law index (0.366), $\dot{\gamma}$ is the local, spatially varying shear rate. OpenFOAM additionally requires minimum and maximum values of the viscosity to be prescribed, which were set to $5 \times10^{-4}$ Pa· s and $5 \times10^{-2}$ Pa· s. All parameter values were based on published rheological measurements of digesta [20]. The density of whole digesta and water were assumed to be 1000 kg m$^{-3}$.

The vorticity field was computed using the OpenFOAM post-processing tool and the $z$-component of the vorticity ($\omega_z$) was integrated over the corresponding PIV region (Figure 4) and normalised with respect to the total area ($A$) to compute the area normalised vorticity or circulation ($\Gamma$) at a given time:

$$\Gamma(t) = \frac{1}{A} \iint_S \omega_z(t)\, dx\, dy. \tag{4}$$

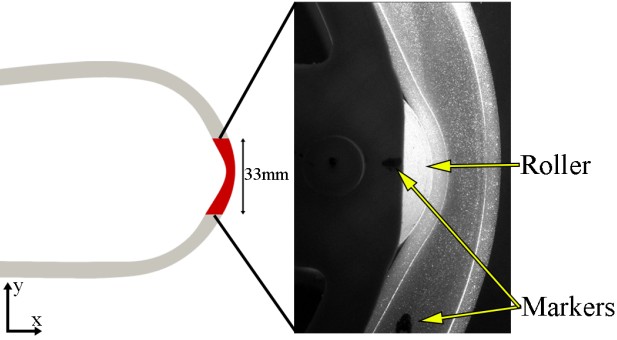

**Figure 4.** PIV region of interest (red surface) within the overall CFD geometry and raw image captured by camera one during PIV study. The markers used for the spatial synchronisation between PIV and deformation images are also indicated.

The circulation within this region provides an integral quantity to compute a relative root mean squared error ($rRMSE$) between the experimental and numerical results (Equation (5)).

$$rRMSE = \left( \frac{1}{\Gamma^m_{max} - \Gamma^m_{min}} \sqrt{\frac{\sum_{i=1}^{N}(\Gamma^e_i - \Gamma^m_i)^2}{N}} \right) \times 100 \tag{5}$$

where $\Gamma^e_i$ and $\Gamma^m_i$ are the PIV and CFD circulations, respectively, at the $i^{\text{th}}$ timestep and $N = 187$ is the total number of timesteps during which the roller moved across the PIV region of interest (1.86 s), $\Gamma^m_{max}$ and $\Gamma^m_{min}$ are the maximum and minimum CFD circulation values over this period.

Spatial synchronisation between the PIV and deformation images was conducted by overlaying the PIV images with the images obtained from the first deformation study (front view) and using the relative location between two markers placed on the roller and the tube (Figure 4). Time-synchronisation was determined from a cross-correlation of PIV and CFD circulation.

## 3. Results

### 3.1. Tube Deformation Patterns

The results of the 3D reconstruction process as the peristaltic contraction propagates along the C-shaped tube are shown in Figure 5. The resting tube geometry (Figure 5A) highlights the slightly stretched state of the tube in the regions where there is contact with the tube guide, as evident from the cross-section shown in the middle panel. Because of the small cut-out region on either side of the roller attachment (Figure 1B), there is a region close to the roller where the tube-guide does not make contact with the tube and this produces a cross-section, which is more circular than the resting state (Figure 5B). Once the roller makes contact with the tube, the deformed state is shown in Figure 5C, where the tube undergoes maximal contraction along the $x$-axis and maximal expansion along the $z$-axis.

### 3.2. PIV Results

The results of the PIV analysis conducted in PIVLab are presented in Figure 6 where the cross-correlation between consecutive images produces the velocity field (Figure 6C) at a given time. Underneath the roller, fluid moves in the opposite direction to the roller (5–10 mm/s), and in regions far away from the roller the fluid is moving with relatively low velocities (1–5 mm/s).

### 3.3. Repeatability of Contraction Patterns

In order to assess the variability in peristaltic contractions between cycles, three full cycles were imaged from the front (camera 1). The fully contracted radius in the $x$-direction was tracked over the three cycles and there were negligible differences in the deformation patterns between these cycles (Figure 7a). PIV measurements were also made for three cycles and Figure 7b) shows the velocity vectors at the time point when the tube is maximally contracted. Flow patterns and velocity magnitudes were very similar between the cycles. Therefore, the tube deformation and flow were considered to be repeatable across cycles.

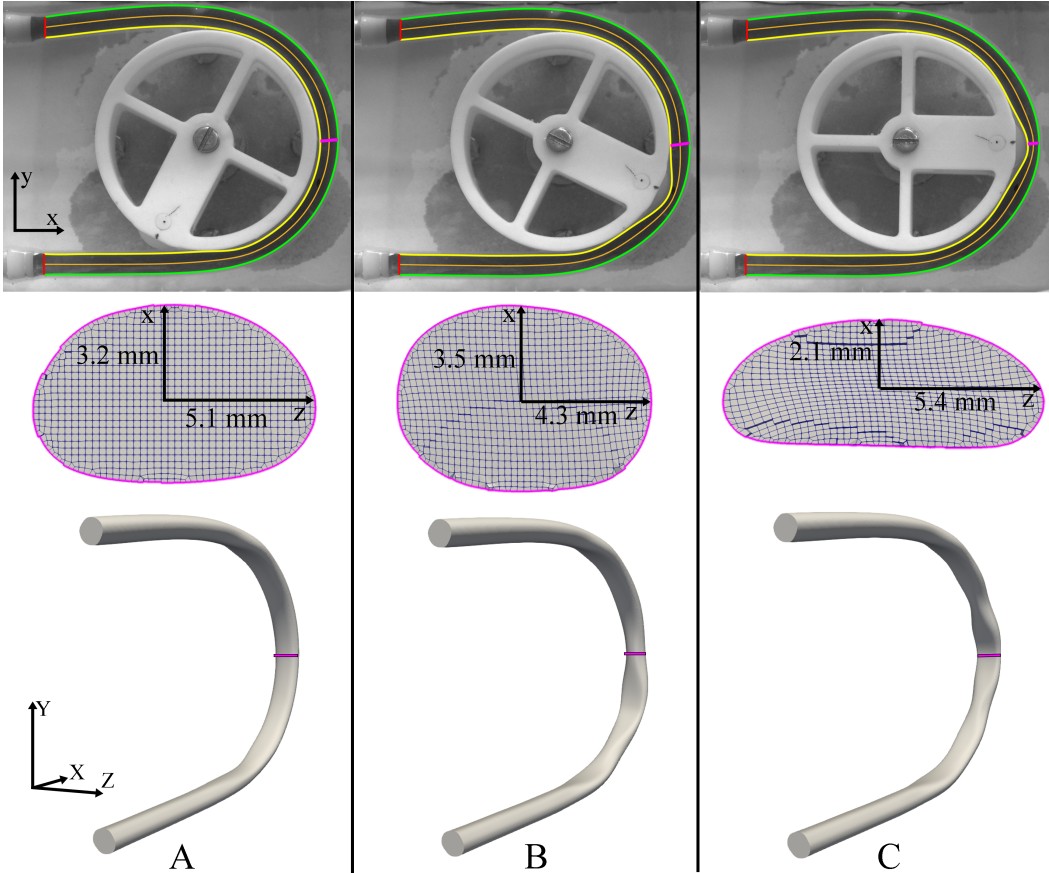

**Figure 5.** 3D Reconstruction of the inner tube walls at three different time-points; (**A**) resting state, (**B**) tube not in contact with the guide, and (**C**) tube is fully deformed. Top panel—inner boundary, outer boundary, and centerline of tube imaged from the front (camera 1). Middle panel—CFD mesh cross-sections (at the position shown by the magenta line) imaged from the bottom (camera 2). Bottom panel—reconstructed 3D geometry at the corresponding instants.

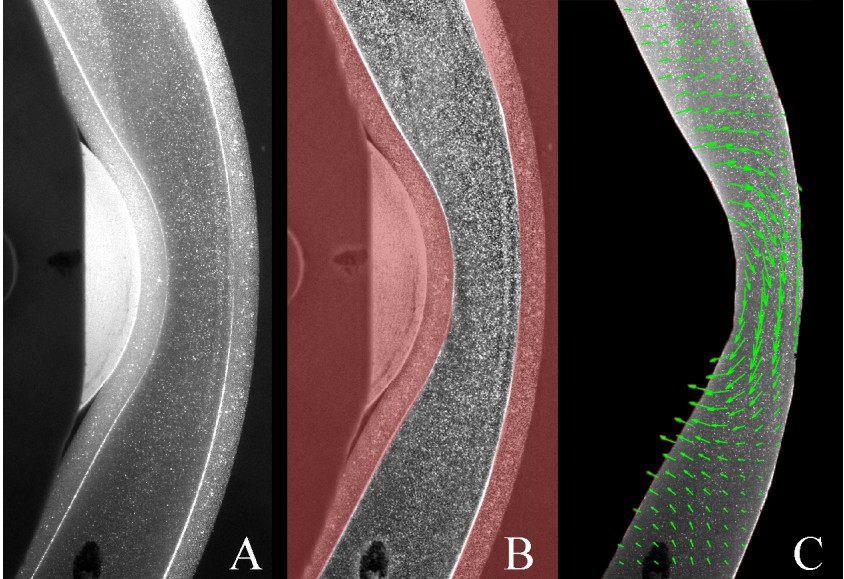

**Figure 6.** PIV analysis; (**A**) raw PIV image, (**B**) masked (region in red) and pre-processed image and (**C**) velocity vectors computed by PIVLab.

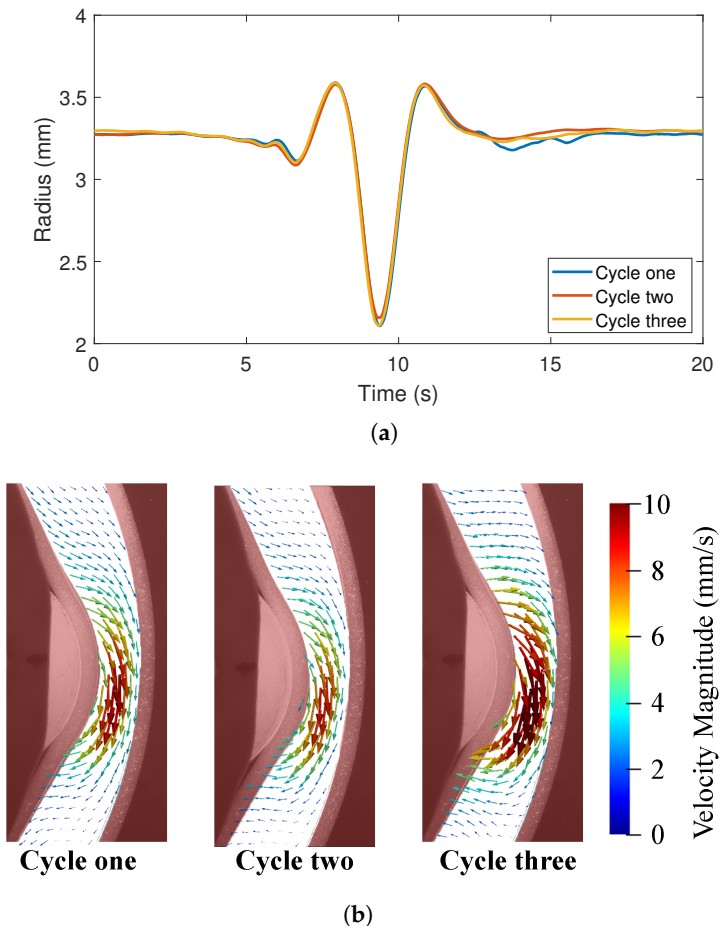

**Figure 7.** Repeatability of experimental results; (**a**) Radius changes along the *x*-axis at the section located in Figure 5C for three deformation cycles and (**b**) PIV velocity vectors at the same point in time of three cycles.

### 3.4. PIV—CFD Comparison

The CFD results from the second cycle were compared to the PIV results from the second cycle and five time points were chosen to show the roller at different locations in the imaging region. Velocity vector fields show qualitative agreement between PIV and CFD results (Figure 8). In both cases, retropulsive flow is seen at all time points, i.e., the flow is opposite to the direction of travel of the roller. The flow field rapidly decays away from the roller and fluid is close to stationary about 1 diameter away from the location of maximum contraction in the PIV images. Decay is slower in the CFD simulations.

Differences between PIV and CFD in the vector directions can be observed at T0 and T2 in regions far away from the roller. At T5, the CFD velocity is substantially lower than the PIV underneath the roller and higher at the lower boundary of the imaging region. Velocity magnitudes observed within the PIV results are typically greater, specifically at the later three time-points where $\overline{U}$ ranges from 0 to 6 mm s$^{-1}$ and 0 to 10 mm s$^{-1}$ in CFD and PIV results, accordingly.

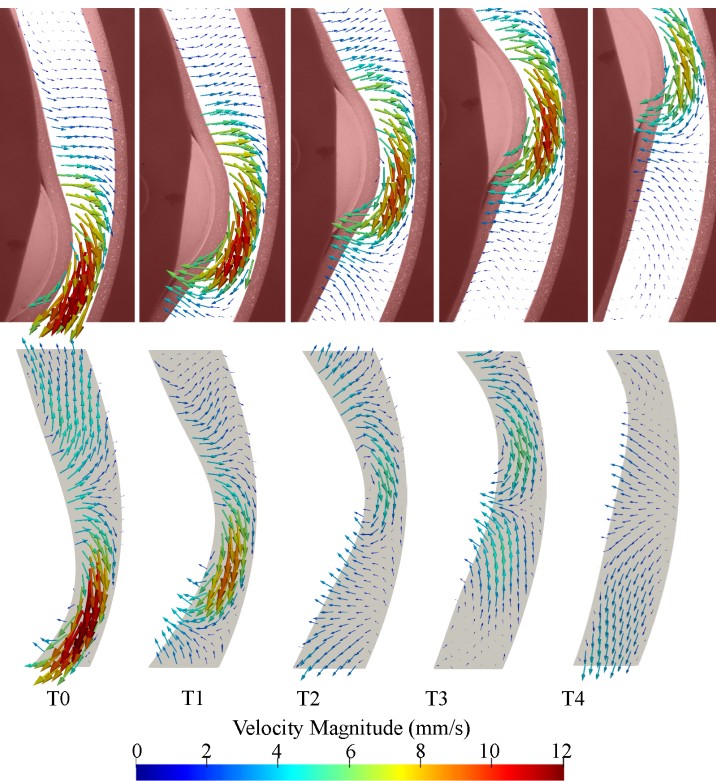

**Figure 8.** Velocity vector comparison between PIV (top panel) and CFD (bottom panel) as the peristaltic contraction moves through the imaging. Snapshots are equally spaced with an interval of 0.4 s. T0 corresponds to 28.77 s relative to the start of the motion, i.e., 8.77 s after the start of the second cycle.

Similar vorticity patterns were observed for the CFD and PIV results with a region of clockwise rotation directly underneath the roller and a region of anticlockwise rotation on the opposite side of the tube (Figure 9). In regions near the top and bottom of the image, the velocity gradients near the walls are not adequately resolved in the PIV results due to the limited spatial resolution of the analysis windows. The regions of high vorticity underneath the roller suggests regions of increased shear present underneath the peristaltic contraction, which is characteristic of the physiological flow patterns within the duodenum.

The circulation (Equation (4)) in the region of interest (Figure 10) was directly compared between experimental and numerical results as a quantitative metric that is relatively insensitive to the noise in the PIV measurements. An increasing trend in the circulation was observed for both PIV and CFD results within this section of the cycle, with the circulation ranging between $-0.33$ and $0.11$. The relative root mean squared error (Equation (5)) was 22% over one cycle. This error in the circulation over one cycle provides a measure of the total spatial and temporal difference between the experimental and numerical results.

Taken together, the agreement between the PIV and CFD-computed velocity, vorticity, and circulation fields indicate that the computational methods can reproduce the features of the physical flow.

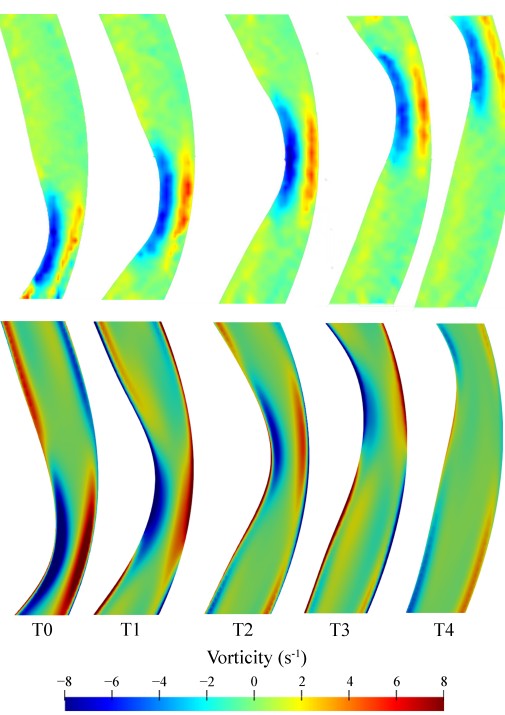

**Figure 9.** Vorticity comparison between PIV (top panel) and CFD (bottom) results as the peristaltic contraction moves along the region of interest. All instants are equally spaced by 0.4 s with T0 corresponding to 28.77 s relative to the start of the motion, i.e., 8.77 s after the start of the second cycle.

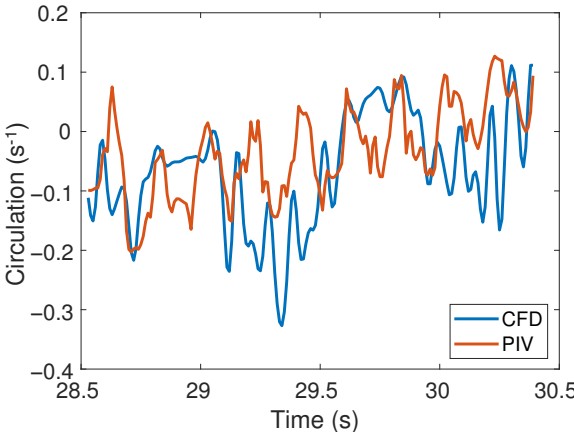

**Figure 10.** Circulation ($\Gamma$) as the roller moves along the middle right-side of the tube for CFD and PIV.

### 3.5. CFD Simulations of Different Fluids

The CFD model was used to examine the effect of varying fluid properties on the flow patterns. Simulations were carried out with properties corresponding to water, glycerol-water mixture and whole digesta. Velocity contours at the maximally contracted cross-section (corresponding to Figure 2C) were computed. Similar velocity distributions were observed in the three fluids (Figure 11a) with differences in the velocity profile near the walls (Figure 11b). As expected, the $y$-component of velocity $U_y$ was the largest. $U_y$ is small but positive close to the wall, and becomes negative away from the wall. The mean flow is therefore opposite to the direction of wave propagation. Higher velocity gradients (higher shear rates) are observed near the walls for water. The mean values of $U_y$ for water, glycerol, and digesta were 4.4, 4.6, and 3.9 mm/s, respectively. The corresponding Reynolds numbers (based on the hydraulic diameter $D = 7.1$ mm) are 39, 3.7, and 0.06–0.6. The relatively low Reynolds number of $Re < 1$ observed with digesta is characteristic of

the flow regime found within the intestines. The Dean number $De = Re\sqrt{\frac{D}{2R_c}}$ for water, glycerol, and digesta was, respectively, 12, 1.1, and 0.018–0.18 based on the hydraulic diameter $D = 7.1$ mm and radius of curvature $R_c = 40$ mm. These relatively low Dean numbers $De < 40$ suggest that secondary flow vortices are not present.

The $x$- and $z$ components of the velocity are roughly similar in magnitude and significantly smaller than the $y$-component. The contour plots (Figure 11a) show at the instant of maximal contraction, fluid is forced towards the centre of the tube along the $z$-direction and towards the walls in the $x$-direction. Thus, there is an overall churning motion that promotes mixing as the peristaltic wave moves through the domain. This pattern persists over the physiologically relevant Reynolds number range (0.06–39). In contrast to the results for $U_y$, regions with non-zero $U_x$ and $U_z$ become increasingly confined to the boundaries at lower $Re$.

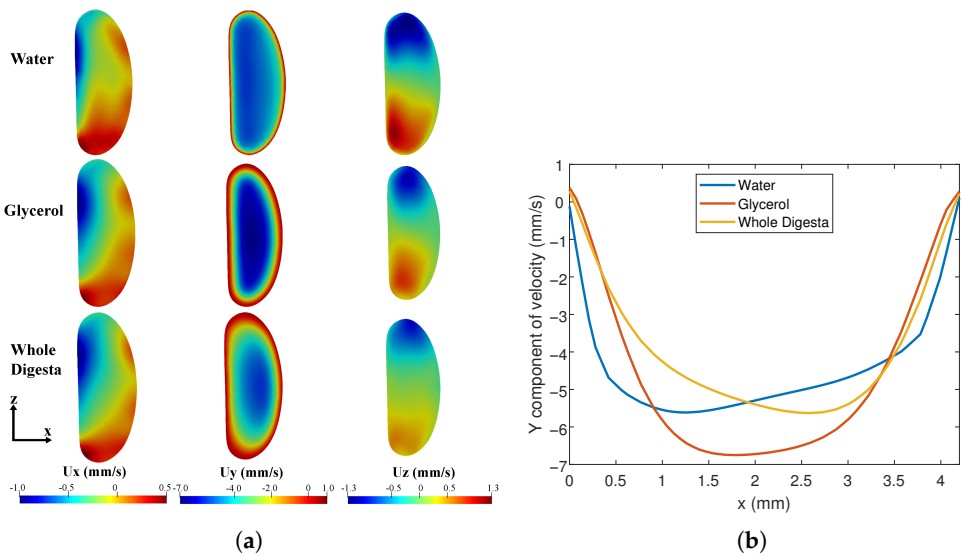

**Figure 11.** Flow comparison for fluids with different viscosities. (**a**) Velocity contours at the cross-section indicated in Figure 5 when the tube is maximally contracted for different fluids; (**b**) Axial flow profile for different fluids.

## 4. Discussion and Conclusions

In this study, a benchtop setup was constructed to simulate peristaltic contractions within a C-shaped tubular geometry representing the duodenum. PIV was used to measure the flow patterns over time in the region of the tube undergoing the greatest contraction. The three-dimensional deformation of the tube was imaged and used to develop a CFD model of the peristaltic flow. The CFD model reproduced the structure of the velocity and vorticity fields observed in the experiments. Peak Reynolds numbers less than 10 were observed for the viscous fluids, characteristic of duodenal contents in both experimental and numerical results. CFD simulations indicated that peristaltic contractions give rise to a churning flow at the location of greatest contraction, which may enhance mixing.

To the best of our knowledge, this is the first study to image peristaltic flow in a curved tube. No other study has directly imaged boundary deformations to construct a CFD model of the experiments. Previous experimental studies of gastrointestinal flows have used a rectangular channel [7] or a straight tube [8]. Similar to those studies, retropulsive flow opposite to the direction of the peristaltic wave was observed at the location of maximum contraction. CFD simulations showed that this feature persists when the fluid viscosity is varied over two orders of magnitude, suggesting that the feature is not sensitive to the geometry or fluid viscosity. This reversed flow is also physiologically significant, as it can potentially improve the mixing of chyme by the sloshing motion. The residence time of chyme is also increased by these fluid motions, thereby allowing for greater nutrient absorption rates.

Comparisons of the velocity and vorticity fields between the PIV and CFD indicated that the CFD techniques capture the essential features of the experimental system. A quantitative comparison of the circulation showed a difference of 22% between the PIV and CFD. CFD estimates of the maximum velocity were lower than PIV measurements and differences in flow fields were observed at some points (Figure 8).

One of the main factors for the discrepancy between the numerical and experimental results is the 3D deformation pattern prescribed for the CFD model. The 3D temporal reconstruction of the geometry involved one main simplification where the $x - z$ plane (out of plane of the peristaltic contraction) deformation was evaluated at a single location. Cross-sectional changes measured at one location were applied along the length of the whole tube based on the roller position obtained from the front view (camera 1). PIV and deformation imaging studies were conducted as separate experiments, which could have slight variations in the experimental conditions and therefore the resulting deformation patterns. This is less likely since the deformation patterns and flow fields over multiple cycles showed little variation (Figure 7). Deviations in the alignment of the 2D laser sheet along the center of the $x - y$ plane for the PIV measurements was another possible factor contributing to the differences.

Overall, the results verify the numerical techniques used to simulate flow driven by prescribed, large amplitude peristaltic boundary deformations. This provides confidence that physiologically realistic in silico models of duodenal flow and mixing can be constructed using anatomically realistic geometries with prescribed boundary deformation. Such models will be useful for understanding the effect of digesta properties, anatomical variations, and peristaltic contraction patterns on mixing and transport in the duodenum in health and disease.

**Author Contributions:** Conceptualization, N.P., J.E.C., L.K.C. and V.S.; methodology, N.P., J.E.C., L.K.C. and V.S.; software, N.P.; validation, N.P., J.E.C. and V.S.; formal analysis, N.P.; investigation, N.P.; resources, J.E.C. and L.K.C.; data curation, N.P.; writing—original draft preparation, N.P.; writing—review and editing, N.P., J.E.C., L.K.C. and V.S.; visualization, N.P.; supervision, J.E.C., L.K.C. and V.S.; project administration, J.E.C., L.K.C. and V.S. All authors have read and agreed to the published version of the manuscript.

**Funding:** This work was funded by the Riddet Institute and a University of Auckland Doctoral scholarship.

**Data Availability Statement:** Data available in a publicly accessible repository The data presented in this study are openly available in FigShare at https://doi.org/10.17608/k6.auckland.c.5792396.v1, accessed on 14 December 2021.

**Acknowledgments:** The authors would like to thank the University of Auckland Thermofluids laboratory technicians, Alan Eaton and Martin Ryder, for helping set up the experiments.

**Conflicts of Interest:** The authors declare no conflict of interest.

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
