# Peer review of "Experimental and Computational Studies of Peristaltic Flow in a Duodenal Model"

_fluids, doi:10.3390/fluids7010040_

Round 1
Reviewer 1 Report
In this article peristaltic flow in a C-shaped compliant tube representing the first section of the duodenum.
PIV has been conducted and CFD simulations have been carried out followed by a verification of the CFD technique.
The results are unique, the paper is well written and clearly represents an advance in this field.
The work has been conducted with immense care for which I congratulate the authors.
I can recommend this manuscript for publication.
I have some minor concerns that I would like the authors to address.
Minor Concerns:
1. The error of 22% between experiments and CFD is quite alarming and should be better quantified.
When I look at the figure 7, it appears to me that the deviations in CFD from experiments are localized to certain regions.
The authors should either mark these regions or even better draw some kind of scatter plots that show the areas where there is the most deviation.
A 22% error, if it were present in all the spatial locations would render the study useless as it is a very high error.
2. It is good that a mesh convergence study was carried out and there was only 3% deviation.
However these results should be reported in the form of a plot as it will only add to the value of the paper.
3. In the discussion the authors may add a couple of references relating or associating these results with the physiologic significance.
Author Response
We have revised the manuscript to address the reviewer’s comments. Reviewer comments are listed below with our responses in italics.
The error of 22% between experiments and CFD is quite alarming and should be better quantified. When I look at the figure 7, it appears to me that the deviations in CFD from experiments are localized  to certain regions. The authors should either mark these regions or even better draw some kind of scatter plots that show  the areas where there is the most deviation. A 22% error, if it were present in all the spatial locations would render the study useless as it is a very high error.
The difference of 22% between the PIV and CFD results is an integral quantity over both space and time. Mathematical details of this relative error metric are now included in the text (Equation 5 and lines 228 - 235). A new figure showing the circulation over time as the roller passes over the region of interest is provided (Figure 10). Reasons for the difference between PIV and CFD results are discussed in lines 354-365. In essence, we expect that the principal cause is the difference between the actual and prescribed tube deformation. However, we note that the simulations capture the main spatial and temporal features of the PIV results as seen in Figures 8 and 9 (Figure 7 and 8 in the original version).
It is good that a mesh convergence study was carried out and there was only 3% deviation. However these results should be reported in the form of a plot as it will only add to the value of the paper
Results from mesh convergence study now shown in Figure 3.
Reviewer 2 Report
Review report
Title: Experimental and Computational Studies of Peristaltic Flow in a Duodenal Model
I have gone through the article carefully and found it interesting and innovative in fluid dynamics.
The paper addresses the peristaltic flow in a C-shaped compliant tube representing the first section of the small intestine, the duodenum. The topic is interesting on physical grounds and is presented well in detail. However to my concern if it is a mathematical analysis its mathematical description is poor. Authors describe compliant tube, no slip condition etc but did not add the relevant equations. Also the particular term added for duodenum should be added. Moreover, the solution methodology is not given. Therefore, I suggest authors to revise the submission before it is accepted for publication in the journal.

Author Response
We have revised the manuscript to address the reviewer’s comments. Reviewer comments are listed below with our responses in italics.
Authors describe compliant tube, no slip condition etc but did not add the relevant equations. Also the particular term added for duodenum should be added.
The no slip condition used here expresses the fact that the fluid velocity at the wall is equal to the velocity of the moving wall itself. These details are provided in Sec 2.4 (lines 175-176) with the corresponding mathematical equation.
the solution methodology is not given
We use an off-the-shelf, open source, finite volume based CFD library OpenFOAM for the numerical simulations as stated in Sec 2.4 (lines 169 - 170). Solution methodology requires the specification of a mesh describing the computational domain, prescription of the boundary conditions and selection of appropriate solvers and associated parameters. These details are described in Sec 2.4.
Reviewer 3 Report
A C-shaped compliant tube representing a sectional small intestine was investigated using PIV and CFD. This is a very interesting study and several outcomes have been postulated that deserve merit. Although simplified to a roller action, has been able to neatly mimic the peristalsis. Recent work on ureter peristalis in Computer Methods and Programs in Biomedicine (Computational flow analysis of a single peristaltic wave propagation in the ureter)could be referred to include the fluid structure interaction study using CFD).
A few clarifications:
Line 77-78: "None of the aforementioned in silico studies (including ours) have validated the numerical results". In the next line you specifically state that the objective of the present study is to validate the numerical findings. This looks contrary to me. Please modify the sentences.
The limitation of this study is that, the CFD model does not take into account the silicone properties of the experimental model which might make the difference in the outcomes generated in the study. The authors can provide suitable justification and may include this as part of the future scope.
Additionally the clinical significance of the findings may be reported. The abstract speaks about the reverse flow underneath the roller. This is important finding particular in the digesta. May need to relate to the physiological process. Recent work on ureter peristalis in Computer Methods and Programs in Biomedicine (Computational flow analysis of a single peristaltic wave propagation in the ureter)could be referred to some of the identical discussions.
Author Response
We have revised the manuscript to address the reviewer’s comments. Reviewer comments are listed below with our responses in italics.
Line 77-78: "None of the aforementioned in silico studies (including ours) have validated the numerical results". In the next line you specifically state that the objective of the present study is to validate the numerical findings. This looks contrary to me. Please modify the sentences.
We have modified the text to make it clear that our previous work (Ref 14) did not include any validation of the numerical results. The text now reads: “None of the aforementioned in silico studies (including our previous work ) have validated the numerical results“
The limitation of this study is that the CFD model does not take into account the silicone properties of the experimental model which might make the difference in the outcomes generated in the study. The authors can provide suitable justification and may include this as part of the future scope.
In our approach the motion of the solid boundary is prescribed using the deformation imaging study and the velocity and pressure fields in the fluid are solved for. As a result the mechanical properties of the compliant tube do not need to be specified for the simulations and do not affect the results. Of course, the fluid pressure field will influence the tube deformation, which will then affect the fluid flow and pressure. However, by prescribing the actual boundary motion, the need to consider fluid-structure interaction is circumvented.
Additionally the clinical significance of the findings may be reported. The abstract speaks about the reverse flow underneath the roller. This is important finding particular in the digesta. May need to relate to the physiological process.
We have addressed the similarity in fluid flow patterns (reversed flow) observed in previous experimental studies such as the gastric flow simulator [7] and previous numerical works also display similar flow patterns under peristalsis (lines 338-344). The physiological relevance of the reversed flow is now addressed in lines 346-349.